



# Hybrid Water Adsorption and Solubility Partitioning for Aerosol Hygroscopicity and Droplet Growth

Kanishk Gohil[1], Chun-Ning Mao[1], Dewansh Rastogi[1], Chao Peng[3,4], Mingjin Tang[3,4,5], and Akua Asa-Awuku[1,2]

[1]Department of Chemical and Biomolecular Engineering, University of Maryland, College Park, MD 20742 USA
[2]Department of Chemistry and Biochemistry, University of Maryland, College Park, MD 20742 USA
[3]State Key Laboratory of Organic Geochemistry, Guangdong Key Laboratory of Environmental Protection and Resources Utilization, and Guangdong-Hong Kong-Macao Joint Laboratory for Environmental Pollution and Control, Guangzhou Institute of Geochemistry, Chinese Academy of Sciences, Guangzhou 510640, China
[4]CAS Center for Excellence in Deep Earth Science, Guangzhou 510640, China
[5]University of Chinese Academy of Sciences, Beijing 100049, China

**Correspondence:** Akua Asa-Awuku (asaawuku@umd.edu)

**Abstract.** In this work, we studied the Cloud Condensation Nuclei (CCN) activity and subsaturated droplet growth of phthalic acid (PTA), isophthalic acid, (IPTA) and terephthalic acid (TPTA), significant benzene polycarboxylic acids and structural isomers found in the atmosphere. Köhler Theory can be effectively applied for hygroscopicity analysis of PTA due to its higher aqueous solubility compared to IPTA and TPTA. As with other hygroscopicity studies of partially water-soluble and

effectively water insoluble species, the supersaturated and subsaturated hygroscopicity derived from (KT) principles do not agree. To address the disparities in the sub- and supersaturated droplet growth, we developed a new analytical framework called the Hybrid Activity Model (HAM). HAM incorporates the aqueous solubility of a solute within an adsorption-based activation framework. Frenkel-Halsey-Hill (FHH)-Adsorption Theory (FHH-AT) was combined with the aqueous solubility of the compound to develop HAM. Analysis from HAM was validated using laboratory measurements of pure PTA, IPTA,

TPTA and PTA-IPTA internal mixtures. Furthermore, the results generated using HAM were tested against traditional KT and FHHAT to compare their water uptake predictive capabilities. A single-hygroscopicity parameter was also developed based on the HAM framework. Results show that the HAM based hygroscopicity parameter based can successfully simulate the water uptake behavior of the pure and internally mixed samples. Results indicate that the HAM framework may be applied to atmospheric aerosols of varying chemical structures and aqueous solubility.

## 1 Introduction

Aerosols can affect the global radiative balance and climate by either absorption and scattering of radiation (direct effect of aerosols), or by acting as Cloud Condensation Nuclei (CCN) resulting in cloud formation (indirect effect of aerosols).



While the direct effect is well-studied and understood, the indirect effect is still the most significant source of uncertainties
in climate forcing. This is primarily attributed to the poor understanding of the CCN activity and hygroscopic properties of
organic aerosols (Talley et al. (2013)). Organic aerosols are ubiquitous in the atmosphere. They contribute significantly to the
atmospheric aerosol mass burden and account for 20-90% of total tropospheric fine aerosol mass (Kanakidou et al. (2005)).
Furthermore, organic aerosols can mix with other organic and inorganic species in the atmosphere to modify their the CCN
activity and hygroscopic properties (e.g., but not limited to Schill et al. (2015); Vu et al. (2019); Gácita et al. (2017); Su et al.
(2010); Padró et al. (2012); Baustian et al. (2012); Fofie et al. (2018)). Consequently, the CCN activity of organic aerosols
needs to be well-characterized to reduce uncertainties in the climate forcing due to indirect effect of aerosols.

Much of the CCN related research focuses on highly water-soluble and sparingly water-soluble compounds (e.g., but not
limited to Jing et al. (2018); Samy et al. (2010); Asa-Awuku et al. (2010); Taylor et al. (2017)). For such compounds, Köhler
Theory (KT) is traditionally applied to study their CCN activity and predict their hygroscopic properties. KT explains droplet
growth by combining the water activity described using Raoult's law (solute effect) with the Kelvin effect (curvature effect)
(Köhler (1936)). KT is applied under the assumptions of infinite and spontaneous water solubility of the solute and an infinitely
dilute water droplet solution (Asa-Awuku et al. (2010); Hartz et al. (2006); Kreidenweis and Asa-Awuku (2014); Barati et al.
(2019); Dawson et al. (2020)). That is, if the aerosol instantaneously disassociates in water, traditional KT aptly explains the
droplet growth driven by molar volume and droplet surface tension (Köhler (1936); Sullivan et al. (2009); Giordano et al.
(2015)). These assumptions work well for many aerosols that are highly soluble ($> 10^{-1}$ m$^3$ solute m$^{-3}$ water; Petters and
Kreidenweis (2007)) that form thermodynamically ideal solutions in water. Moreover, the water uptake characteristics of such
highly water-soluble compounds can be predicted with a single KT hygroscopicity parameter ($\kappa$) (Petters and Kreidenweis
(2007)). The $\kappa$ parameter derived in this way is defined as the "intrinsic $\kappa$" of the aerosol.

However, there is an abundance of partially and effectively water insoluble organic compounds in the atmosphere. The
CCN activity of such limited water solubility compounds has been predicted by incorporating the compound solubility in
traditional KT (Hartz et al. (2006); Petters and Kreidenweis (2008)). Riipinen et al. (2015) prescribed a "solubility partitioning"
framework using the traditional KT for CCN analysis of pure and internally mixed aerosols in a large range of aqueous
solubility. Furthermore, a modified $\kappa$ parameter accounting for the water solubility of the aerosols can also be derived based on
this solubility-modified KT framework (Petters and Kreidenweis (2008); Sullivan et al. (2009); Nakao (2017)). This modified
hygroscopicity varies over the course of droplet growth and is dependent on the droplet size. Despite the modifications to
traditional KT, differences have been observed between the experimental $\kappa$ with either the intrinsic or solubility modified
$\kappa$ of the aerosol (Sullivan et al. (2009); Kumar et al. (2009b)). Specifically, aerosols with solubility $< 5 \times 10^{-4}$ m$^3$ m$^{-3}$ are
"effectively insoluble" (Petters and Kreidenweis (2008)) and do not agree with the water uptake predictions using either
traditional or solubility-modified KT.

Droplet growth can be explained for the effectively insoluble organic compounds using a water adsorption framework.
CCN activity from adsorption can be modeled by combining the water activity from an adsorption isotherm with the Kelvin
effect (e.g., but not limited to Kumar et al. (2009a); Kumar et al. (2009b); Kumar et al. (2011a); Kumar et al. (2011b); Rahman
and Al-Abadleh (2018); Malek et al. (2022); Tang et al. (2016); Henson (2007); Goodman et al. (2001); Hatch et al. (2012)).



One such mathematical formulation accounts for adsorption using the Frenkel-Halsey-Hill (FHH) isotherm (Sorjamaa and
Laaksonen (2007)). The FHH isotherm consists of 2 empirical parameters denoted as $A_{FHH}$ and $B_{FHH}$. $A_{FHH}$ explains the
interaction of the first adsorbed water layer and the particle surface, while $B_{FHH}$ explains the interaction between subsequently
adsorbed water layers and the particle. The FHH isotherm combined with the Kelvin effect provides the FHH-Adsorption
Theory (FHH-AT) for CCN activity analysis. Parameters specific to a given aerosol species can be experimentally determined
by fitting FHH-AT through their CCN activity measurements. Studies have so far explored the application of the FHH-AT
for CCN analysis of several water-insoluble compounds (e.g., but not limited to Kumar et al. (2009a); Kumar et al. (2009b);
Kumar et al. (2011a); Kumar et al. (2011b); Hatch et al. (2014); Hatch et al. (2019); Dalirian et al. (2018); Laaksonen et al.
(2016); Laaksonen et al. (2020)). FHH-AT consists of 2 empirical parameters as opposed to a single $\kappa$ parameter in traditional
or modified KT. Additionally, an important assumption in FHH-AT and other similar adsorption models is that the aerosols are
treated as completely water insoluble. Only recently, in a companion paper (Mao et al. (2022), *submitted*) has FHH-AT been
shown to work for insoluble particles with water-soluble and molecular level functionalized surfaces. Thus, there now exists a
transitional regime from a soluble to water-insoluble models to correctly describe droplet growth.

The following paper probes several aspects of water uptake to develop a comprehensive model to describe droplet for-
mation of effectively water insoluble to partially soluble organics. Specifically, a new CCN activity model is developed by
combining the components of the solubility modified KT with the FHH isotherm. This work is a companion and extension
to the single-parameter framework developed in Mao et al (2022) (*submitted*). Throughout this paper, this model will be re-
ferred to as the Hybrid Activity Model (HAM). Within the HAM framework, the aerosol particles are treated as completely
water insoluble at the start of the droplet growth process. The particle continues to fractionally dissolve into the aqueous phase
as droplet growth progresses. While the dissolved fraction of the aerosol contributes to droplet growth via Raoult's law, the
undissolved fraction contributes to droplet growth via adsorption of water on the surface. Furthermore, this work discusses
the development of a single $\kappa$ parameter based on HAM to represent the effect of aqueous solubility on droplet growth for a
compound that would be otherwise treated as effectively water insoluble.

The development and application of HAM is explained in this paper using the experimental droplet growth measurements
of 3 low water solubility structural isomers of benzene di-carboxylic acid – Phthalic acid (PTA), Isophthalic acid (IPTA) and
Terephthalic acid (TPTA). PTA, IPTA and TPTA are among some of the significant benzene polycarboxylic acids detected in
the atmosphere (Haque et al. (2019); Meng et al. (2018); Liu et al. (2019); Yassine et al. (2020); Fu et al. (2009); Kanellopoulos
et al. (2021); Singh et al. (2017); Kunwar et al. (2019)). PTA and its isomers are known to be tracers of benzanthracene,
naphthalene-1 and methylnaphthalene-1, prominent emissions from combustion (Kleindienst et al. (2012); Al-Naiema et al.
(2020); He et al. (2018)). PTA is also a byproduct of pre-ozonation of fulvic acid, another significant marker of biomass burning
emissions (Zhong et al. (2017b); Zhong et al. (2017a)). IPTA and TPTA are also predominantly produced from biomass burning
and emissions of automobile exhausts (Kawamura and Kaplan (1987); Mkoma and Kawamura (2013); Balla et al. (2018); Al-
Naiema and Stone (2017)).

The hygroscopic properties of PTA, IPTA and TPTA have been studied in the past (e.g., but not limited to Petters and
Kreidenweis (2007); Hartz et al. (2006); Wang et al. (2021)). However, a comprehensive comparison and discussion of the





effects of structural isomers on the droplet growth of benzene-dicarboxylic acids does not exist. Vapor sorption measurements

of bulk PTA indicate hygroscopic growth at high ambient relative humidity (> 90% RH) (Wang et al. (2021)). Hämeri et al. (2002) used Tandem Differential Mobility Analyzer (TDMA) technology and observed that PTA aerosol did not grow in sub-saturated conditions. Other studies show PTA internal mixtures with inorganics can deliquesce under subsaturated conditions (Jing et al. (2016); Jing et al. (2018)). Furthermore, Hartz et al. (2006) showed that PTA could activate as CCN at 1% super-saturation. The activation was consistent with KT that assumed complete dissolution with no solubility considerations. Petters

and Kreidenweis (2007) report the $\kappa$ = 0.059 and 0.051 for PTA under sub- and supersaturated conditions, respectively. To our knowledge only one other paper has measured droplet growth of IPTA. Hartz et al. (2006) found that IPTA behaves as an insoluble compound and does not obey traditional KT. Few studies have measured hygroscopic properties of TPTA, but not in context of CCN (Diniz et al. (2017); Zhao et al. (2021)). To our knowledge, the application of adsorption models has not been studied for CCN analysis of PTA, IPTA or TPTA.

Overall, HAM is used in this paper to extensively study the hygroscopic properties of PTA, IPTA and TPTA that are not yet cogently known. In addition to the aforementioned pure compounds, the internal mixtures of PTA and IPTA are also studied. The compounds and their mixtures considered in this work are useful and help us understand the efficacy of different CCN models to describe the droplet growth associated with different organic CCN with varying aqueous solubilities. The experimental CCN measurements provide an efficient means to validate the application of the newly developed HAM. In the

following sections, we first describe the experimental setup used in this study to obtain droplet growth data for PTA, IPTA and TPTA and PTA-IPTA internal mixtures. We then describe the theory and formulation of HAM based on KT and FHH-AT, and how it was implemented for droplet growth analysis of aerosols. We subsequently explain the derivation of the single $\kappa$ parameter using the HAM framework, followed by the discussion of results and conclusions of this study.

## 2 Experimental Section

### 110 2.1 Compounds and Aerosol Generation

Phthalic acid (PTA, 1,2 – benzenedicarboxylic acid, >99.5%, Sigma-Aldrich®) and terephthalic acid (TPTA, 1,4 – benzenedi-carboxylic acid, 98%, Sigma-Aldrich®) and Isophthalic acid (IPTA, 1,3 – benzenedicarboxylic acid, >99%, Fisher Scientific®) were used as representative compounds for the aromatic acid aerosols (AAAs, hereafter). The physical properties of PTA, IPTA and TPTA are summarized in Table 1. Aqueous solutions of PTA, IPTA and TPTA were formed by mixing 30 mg of acid in 500

ml of ultrapure water (Milli-Q or Millipore®, 18.2 MΩ cm-1). Additionally, 3 internally mixed solutions of PTA and IPTA were also prepared by mixing 30 mg of dry acid mixture in 500 ml ultrapure water. The internally mixed solutions were prepared for 3 different mass fractions of PTA and IPTA (5:1, 1:1 and 1:5 wt/wt). To facilitate the dissolution of solute in aqueous solution, all the solutions were sonicated for 2 hours in a warm water bath maintained at ∼ 40 °C to create a uniform suspension. The solution was subsequently cooled and maintained at 20 °C. Polydisperse aerosols were generated using a Collison Nebulizer

(TSI Atomizer 3076). The wet aerosol particles were then passed through a series of 2 silica gel diffusion dryers (TSI 3062) to remove moisture (to RH < 10%). The dry particles were then classified for supersaturated and subsaturated measurements.



## 2.2 CCNC Experiments for Supersaturated Measurements and Data Analysis

A continuous flow stream-wise thermal gradient Cloud Condensation Nuclei Counter (CCNC, Droplet Measurement Technologies (DMT) (Roberts and Nenes (2005)) - CCN 100) was used for the droplet activation measurements (e.g., but not limited to Engelhart et al. (2008); Moore et al. (2010); Tang et al. (2012); Barati et al. (2019); Vu et al. (2019)) of AAAs in supersaturated conditions. Briefly described here, polydisperse aerosol was generated and dried as described in Sect. 2.1. The electrical mobility aerosol size from 8 nm to 352 nm was measured with an electrostatic classifier (TSI 3936, DMA 3081, and CPC 3776) every 2.25 minutes. The size-selected aerosols exiting the DMA were then split into 2 streams. A Condensation Particle Counter (CPC, TSI 3776) samples the first stream at 0.3 L min$^{-1}$ to measure total dry particle concentration ($C_{CN}$), and the CCNC samples the second stream at 0.5 L min$^{-1}$ and constant supersaturation to measure activated particle (droplet) counts ($C_{CCN}$). A sheath flow rate of 8 L min$^{-1}$ was applied to maintain a sheath-to-sample ratio of 10:1 across the experimental setup. The measurements were repeated 10 times for each supersaturation. Furthermore, the measurements were performed over supersaturations ranging between 0.6% and 1.6%. CCNC supersaturations were calibrated using ammonium sulfate ($(NH_4)_2SO_4$, AS) aerosol (Sigma-Aldrich®, >99.9%). AS data used for CCN calibration is provided in the supplemental information (Section S1).

PyCAT 1.0 (Gohil and Asa-Awuku (2022)) was employed for data processing, analysis, and visualization of the CCN measurements. CCN size-resolved activation curves were generated at a fixed supersaturation ($S$) as ($\frac{C_{CCN}}{C_{CN}}$) across a range of dry particle diameters ($D_{dry}$). The volume equivalent diameters were used to represent particle sizes that were obtained by combining size-resolved particle dynamic shape factor ($\chi$) with measured electrical mobility diameters (see supplemental Fig. S4). Multiple charging errors were removed from the size-resolved activation ratio following a combination of charge correction algorithms from Gunn (1956) and Wiedensohler (1988). Following this, a Boltzmann sigmoidal fit expressed as,

$$y = \frac{(A_1 - A_2)}{1 + e^{(x-x_0)/dx}} - A_2 \tag{1}$$

was applied to the size-resolved activation ratio curve. In Eq. (1), $y$ is the dependent variable $\frac{C_{CCN}}{C_{CN}}$, $A_1$ and $A_2$ are the minimum and maximum of the sigmoid respectively, $dx$ is the slope of the sigmoid, $x_0$ is the inflection point of the sigmoid (generally the midpoint of the sigmoid), and $x$ is the independent variable ($D_{dry}$). The sigmoid fit is typically scaled over a range of 0.0 to 1.0, and so $x_0$ corresponds to the critical dry diameter ($D_{dry,c}$) at the instrument supersaturation and is physically defined as the size at which 50% of all particles are activated.

## 2.3 H-TDMA Experiments for Subsaturated Measurements

A Hygroscopicity Tandem Differential Mobility Analyzer (H-TDMA) measured droplet growth of AAAs in the subsaturated regime. The H-TDMA setup has been previously explained in detail (Rader and McMurry (1986); Cruz and Pandis (2000)) and only a brief description is provided here. Dried polydisperse aerosol were first charged with a Kr-85 bipolar aerosol neutralizer (TSI 3081). Monodisperse charged particles with a dry diameter ($D_{dry}$) were size selected using a Differential Mobility Analyzer (DMA 1). The sample and the sheath flow rates were maintained at 0.3 L min$^{-1}$ and 3.0 L min$^{-1}$ respectively (i.e.,





sheath-to-sample flow ratio = 10:1). The size-selected particles from DMA 1 were then exposed to $95 \pm 0.46\%$ RH using a
nafion humidification membrane (PermaPure M.H series). The humidified aerosol stream was then passed through the second
DMA (DMA 2) that was equilibrated to a constant RH. DMA 2 was coupled with a Condensation Particle Counter (CPC, TSI
3756) and operated in Scanning Mobility Particle Sizer (SMPS) mode. The median wet diameter ($D_{wet}$) of the size-resolved
number concentration of the humidified aerosol stream from DMA 2 was reported. $D_{wet}$ was used as the approximate final
size to which the particles of size $D_{dry}$ would grow under $95 \pm 0.46\%$ RH conditions. The hygroscopic growth factors ($G_f$)
were obtained by taking the ratio of $D_{wet}$ with respective $D_{dry}$,

$$G_f = \frac{D_{wet}}{D_{dry}} \tag{2}$$

The RH of H-TDMA setup was calibrated using ammonium sulfate (see Figure S1; Taylor et al). (($NH_4)_2SO_4$, AS) aerosol
(Sigma-Aldrich ®, >99.9%). Calibration data is found in the supplemental information.

### 2.4  VSA Experiments for Subsaturated Measurements

A vapor sorption analyzer (VSA, TA Instruments New Castle, DE, USA) setup was used for the hygroscopicity measurements
of bulk samples in the subsaturated regime. Mass change in AAAs as a function of RH (5-95%) was measured at 25 °C. The
instrument setup for the VSA has been described in detail in literature (Gu et al. (2017)), and thus, experimental procedure is
briefly explained here. During each experiment, bulk samples were first dried at <1% RH, then the RH was incremented up to
90% with a 10% step and followed by a 5% step from 90 to 95%. A high-precision balance was used in the VSA to measure
the sample mass at different RHs with a stated sensitivity of <0.1 $\mu$g. For every RH, a $\leq$ 0.1% change in the sample mass
was considered as the standard for stabilization. The initial dry mass of AAA samples used in this measurement was typically
around 1.0 mg. For each sample, a minimum of 3 experiments were performed. At every RH, the sample mass ($m$) was
normalized with respect to the initial mass of the dry sample ($m_0$). Subsequently, the mass-based growth factor was calculated
as $\frac{m}{m_0}$.

## 3  Water Uptake and Hygroscopic Theory and Analysis

### 3.1  Köhler Theory (KT)

The equilibrium supersaturation ($S$) can be estimated over a droplet as a function of its size ($D_p$) as,

$$S = a_{w,KT} \cdot \exp\left(\frac{4\sigma_{s/a}M_w}{RT\rho_w D_p}\right) \tag{3}$$

where $a_w$ is the water activity term, $\sigma_{s/a}$ is the droplet surface tension at the interface, $M_w$ and $\rho_w$ are respectively the
molecular weight and density of water, $R$ is the universal gas constant (8.314 J mol$^{-1}$ K$^{-1}$), and $T$ is the temperature. The
water activity is mathematically expressed as $a_{w,KT} = \gamma_w x_w$, where $\gamma_w$ and $x_w$ are the activity coefficient and mole fraction
of water in the droplet, respectively. In traditional Köhler Theory (KT), the water activity is approximated as $a_{w,KT} = x_w$





(Raoult's law), which assumes infinite dilution and complete dissolution of the solute. Furthermore, $\sigma_{s/a}$ is the surface tension of a pure water droplet. The exponential quantity is the Kelvin term that describes the curvature effect. The solute effect and

curvature effect are competing effects that describe droplet growth – the solute effect accounts for the water vapor pressure drop over the droplet due to the aerosol particle, and the curvature effect accounts for the water vapor rise over the droplet due to surface tension reduction.

### 3.2   Frenkel-Halsey-Hill (FHH) Adsorption Theory (FHH-AT)

Traditional KT, with or without the explicit treatment of aerosol solubility, can be effectively applied for highly soluble species.

However, for partially or completely insoluble species Raoult's law is substituted with adsorption isotherms to model water uptake behavior. One such isotherm is the Frenkel-Halsey-Hill (FHH) adsorption isotherm that defines water activity through multilayer water adsorption as a function of relative surface coverage ($\theta$, or the number of adsorbed water monomolecular layers). The FHH isotherm is expressed as (Sorjamaa and Laaksonen (2007)),

$$a_{w,FHH} = \exp(-A_{FHH}\theta^{-B_{FHH}}) \tag{4}$$

where $A_{FHH}$ and $B_{FHH}$ are FHH fit parameters that describe the intermolecular interactions responsible for the adsorption of water on particle surfaces. $A_{FHH}$ describes the interactions between the particle surface and first adsorbed water monolayer. $B_{FHH}$ describes the interactions between successively adsorbed monolayers. $A_{FHH}$ and $B_{FHH}$ regulate the amount of water adsorbed on the particle surface and the radial distance up to which attractive forces can contribute to adsorption of water, respectively. $\theta$ in Eq. (4) is expressed as $\frac{D_p - D_{dry}}{2 \cdot D_w}$ where $D_p$ and $D_{dry}$ have been previously defined, and $D_w$ is the size of

the water molecule. The mathematical representation for the FHH-AT is analogous to traditional KT, and combines the FHH isotherm with the Kelvin term (Sorjamaa and Laaksonen (2007); Kumar et al. (2009a)) such that,

$$S = a_{w,FHH} \cdot \exp\left(\frac{4\sigma_w M_w}{RT\rho_w D_p}\right) \tag{5}$$

The FHH parameters can be empirically determined for any aerosol species from their droplet growth measurements (Kumar et al. (2009a)). For measurements in supersaturated environments, $A_{FHH}$ and $B_{FHH}$ are determined from least square

minimization of the experimental data with the maxima of the FHH-AT equilibrium curves (Kumar et al. (2009a); Kumar et al. (2009b); Kumar et al. (2011a); Kumar et al. (2011b)). A higher value of $A_{FHH}$ implies a higher water adsorption, and a smaller value of $B_{FHH}$ implies stronger attractive forces over larger distances. It has been observed that $B_{FHH}$ has a larger influence on the shape of the adsorption isotherm, and hence strongly drives CCN activation using FHH-AT (Kumar et al. (2009a); Hatch et al. (2019)).

### 210  3.3   Hybrid Activity Model (HAM)

The assumptions of complete aqueous solubility or insolubility associated with KT and FHH-AT, respectively, represent two extreme possibilities of CCN activation and droplet growth. In this work, the two water activities were combined to develop a generalized "hybrid" water activity term. The droplet growth model thus obtained is called the Hybrid Activity Model, or



HAM. The general mathematical representation of HAM is as follows,

$$S = a_{w,HAM} \exp\left(\frac{4\sigma_w M_w}{RT\rho_w D_p}\right) \tag{6}$$

where $a_{w,HAM} = a_{w,KT} \cdot a_{w,FHH}$, and the definitions of $a_{w,KT}$ and $a_{w,FHH}$ are provided in subsections 3.1 and 3.2, respectively.

HAM sandwiches different phases of droplet growth for any given particle in three stages. In stage 1, HAM assumes that a particle suspended in humidified ambient conditions does not dissolve at the start of the activation process (time, t⟶0). That is, droplet growth at t⟶0 occurs entirely due to the adsorption of a water monolayer on the particle surface and can be explained using the FHH isotherm. In stage 1,

$$a_{w,HAM,1} = a_{w,FHH} = \exp\left(-A_{FHH}\theta^{-B_{FHH}}\right) \tag{6a}$$

The FHH parameters $(A_{FHH}, B_{FHH})$ for any given species are determined by fitting the FHH-AT to the experimental data and can be subsequently used in the HAM framework.

Stage 2 begins as the droplet continues to grow and more water accumulates in the aqueous phase. In this stage, the particle starts dissolving and enters the aqueous phase. The fraction of particle mass that dissolves or enters the aqueous phase depends on the solubility of the compound. Moreover, the dissolved fraction of the particle can be estimated at each step of droplet growth using the solubility partitioning concept introduced by Riipinen et al. (2015). Briefly described here – a droplet comprises of a bulk dry (undissolved) phase and an aqueous (dissolved) phase. The bulk phase can be composed of one or more internally mixed species with varying water solubility. This causes the composition and core size of the bulk phase to vary dynamically during droplet growth. The amount of water in the aqueous phase increases as the droplet grows, thereby increasing the concentration of the compounds in the aqueous phase. There is a competition for dissolution between the compounds in the bulk phase which is dependent on their solubilities. Considering a dry particle consisting of $n$ species with limited solubility, the undissolved mass fraction of a species $i$ $(\chi_i)$ during droplet growth is expressed as (Riipinen et al. (2015)),

$$\chi_i = 1 - \frac{\gamma_i \chi_i Y_{i,dry} c_{i,pure} m_w}{m_{i,dry} \sum_i \chi_i Y_{i,dry}} \tag{6a-1}$$

where $\gamma_i$ is the activity coefficient, $c_i$ $(g\ gH_2O^{-1})$ is the solubility of the pure species, $m_w$ is the mass of water in the droplet, $m_{i,dry}$ is the initial mass of the pure species in the dry particle, and $Y_{i,dry}$ is the initial mole fraction of the pure species in the dry particle. Eq. (6a-1) implies that the dissolved mass fraction of the species $i$ in the aqueous phase is given as $1 - \chi_i$. A set of $n$ coupled equations are simultaneously solved to obtain $\chi_i$ for all $n$ species in the mixture. $\chi_i$ is then used to calculate the mole fraction of species $i$ dissolved in the aqueous phase $(x_i)$ at any point during droplet growth. Subsequently, the KT water activity can be given as $a_{w,KT} = x_w = \frac{n_s}{n_s + n_w}$, where $n_s$ and $n_w$ are respectively the number of moles of solute and water in the aqueous phase. In stage 2, the contribution of the dissolved fraction of the compound in the aqueous phase (through Raoult's law) can be combined with the undissolved fraction in the solid phase (through the FHH isotherm) to generate the overall water activity term,

$$a_{w,HAM,2} = a_{w,KT} \cdot a_{w,FHH} = x_w \cdot \exp\left(-A_{FHH}\theta^{-B_{FHH}}\right) \tag{6b}$$





Eq. (6b) highlights the main difference between the models presented by Kumar et al. (2009a) and Riipinen et al. (2015).

Stage 3 begins when the droplet is large enough to accommodate enough water in the aqueous phase and dissolve the particle mass entirely. This point onward, the droplet growth can be explained using traditional KT. In stage 3,

$$a_{w,HAM,3} = a_{w,KT} = x_w \tag{6c}$$

Eq. (6a), (6b) and (6c) were combined to describe the water activity through the three stages of droplet growth in the HAM framework. Thus HAM can effectively estimate the droplet growth across a wide range of aqueous solubilities. The HAM sandwiches 2 extremes represented by fully soluble and fully insoluble behavior in a single framework. Indeed one can consider it to be (H)AMbidextrous and apply the concept to improve upon the single hygroscopicity parameterization ($\kappa$).

### 3.4 Hygroscopicity Parameterization – Single-Hygroscopicity Parameter ($\kappa$)

Commonly, the CCN activity and water uptake tendencies of any given compound is expressed using a single hygroscopicity parameter ($\kappa$). A theoretical $\kappa$ is derived using a simple parameterization of the solute water activity term in the droplet growth model. Additionally, critical dry particle sizes can be combined with their supersaturations to experimentally determine $\kappa$. In the following subsections, the $\kappa$ parameter derived from different models are explained.

### 3.4.1 KT Hygroscopicity

A single hygroscopicity parameter ($\kappa$) has been developed using the KT framework. $\kappa$ can be defined through its effect on the water activity in the droplet as follows,

$$a_w^{-1} = 1 + \kappa \frac{V_s}{V_w} \tag{7}$$

where $V_s$ is the dry particulate (solute) volume, and $V_w$ is the volume of water in the droplet. $\kappa$ obtained from Eq. (7) is a 
parameterized quantity determined from the water activity based on the Raoult's law. Using $\kappa$-based parameterization of $a_w$, Eq. (3) can be modified for any $D_{dry}$ as,

$$S = \left( \frac{D_p^3 - D_{dry}^3}{D_p^3 - (1 - \kappa) \cdot D_{dry}^3} \right) \cdot \exp\left( \frac{4\sigma_w M_w}{RT\rho_w D_p} \right) \tag{8}$$

For a given $D_{dry}$, the droplet size increases as the supersaturation above the droplet surface increases. Supersaturation increases until the point of activation, which is characterised using the critical wet droplet size ($D_{p,c}$). The supersaturation at 
the point of activation along with the corresponding $D_{dry}$ and $D_{p,c}$ depend on the $\kappa$ of the compound. $\kappa$ of any compound in an aqueous phase is difficult to measure, but it can be theoretically approximated using the Raoult's law ($\kappa_{intrinsic}$). The $\kappa_{intrinsic}$ of any species (denoted using a subscript $i$) can be expressed as follows,

$$\kappa_{intrinsic,i} = \frac{\nu \rho_i M_w}{\rho_w M_i} \tag{9}$$

where $\nu$ is the Van't Hoff factor of the compound and is related to its aqueous dissociation, $M_i$ and $M_w$ are the molecular 
weights of the solute $i$ and water, and $\rho_i$ and $\rho_w$ are the density of the solute $i$ and water, respectively. $\kappa_{intrinsic,i}$ defined in



Eq. (9) here is dependent only on solute composition and solvent (water) properties, and is independent of size. $\kappa_{intrinsic}$ of a mixture can be computed using a volume average mixing rule with the Zdanovskii-Stokes-Robinson (ZSR) approximation as follows (Petters and Kreidenweis (2007)),

$$\kappa_{intrinsic} = \sum_i \epsilon_i \kappa_{intrinsic,i} \tag{10}$$

where $\epsilon_i$ is the volume fraction of $i^{th}$ component in the dry particle and $\kappa_{intrinsic,i}$ is the intrinsic hygroscopicity parameter of the $i^{th}$ component. $\epsilon_i$ in an internal mixture of $n$ components is estimated as $\epsilon_i = \frac{m_i/\rho_i}{\sum_i^n m_i/\rho_i}$, where $m_i$ is the mass of the pure component $i$ in the mixture. $\kappa$ in Eq. (10) assumes complete aqueous solublity of the compound or mixture. The hygroscopicity parameterization requires explicit treatment of aqueous solubility for compounds that are inherently insoluble or sparingly soluble but possess water uptake tendencies (Petters and Kreidenweis (2008); Sullivan et al. (2009)). In such cases,

$\kappa$ is mathematically expressed by modifying $\kappa_{intrinsic,i}$ of the mixture components ($\kappa_{solubility}$) as follows,

$$\kappa_{solubility} = \sum_i \epsilon_i \kappa_{intrinsic,i} H(x_i) \tag{11a}$$

$$x_i = \left( \frac{D_p^3}{D_{dry}^3} - 1 \right) \frac{C_i}{\epsilon_i} \tag{11b}$$

$$H(x_i) = \begin{cases} x_i & x_i < 1 \\ 1 & x_i > 1 \end{cases} \tag{11c}$$

where $C_i$ is the water solubility of the $i^{th}$ component of the dry particle (expressed as solute volume per volume of water), $x_i$

is the fraction of $i^{th}$ component dissolved in water, and $H(x_i)$ is the distribution function of the fraction of the $i^{th}$ component dissolved in water. Eq. (11) (a)-(c) determines $\kappa$ as a function of $D_p$. For unknown species with limited water solubility, some range of $D_p$ corresponds to a volume of water which might not be sufficient to dissolve the volume of dry particle. Therefore, experimental droplet growth data is required to determine particle hygroscopicity.

$\kappa$ can also be determined if the supersaturation ($S$) and the critical dry diameter ($D_{dry,c}$) measured at $S$ are experimentally

known. The experimental $\kappa$ derived using KT is denoted as $\kappa_{KT}$ and expressed as follows,

$$\kappa_{KT} = \frac{4\left( \frac{4\sigma_w M_w}{RT\rho_w} \right)^3}{27 D_{dry,c}^3 \log^2(S)} \tag{12}$$

Eq. (12) also incorporates the same set of assumptions as Eq. (7)-(11) – dilute solution, and infinite and complete solubility of the compound.

### 3.4.2 FHH-AT Hygroscopicity

For the FHH-AT, a similar $\kappa$ parameterization as KT can be developed by combining the water activity with the FHH isotherm using Eq. (7) (Mao et al, $submitted$),

$$a_{w,FHH} = \left[ 1 + \kappa_{FHH} \frac{\cdot V_s}{V_w} \right]^{-1} = \exp\left( -A_{FHH} \cdot \theta^{-B_{FHH}} \right) \tag{13}$$





which can be expanded to derive the FHH single hygroscopicity parameter $(\kappa_{FHH})$. The $\kappa_{FHH}$ thus determined depends on the experimental data. The measured $S$ and the corresponding $D_{dry,c}$ can be used to compute the $D_{p,c}$ using Eq. (5) and subsequently used to estimate $\kappa_{FHH}$ as follows,

$$\kappa_{FHH} = \frac{6\theta D_w}{D_{dry,c}} \cdot \left[ \frac{1}{\exp(-A_{FHH}\theta^{-B_{FHH}})} - 1 \right] \implies f(D_{dry,c}, D_{p,c}) \tag{14}$$

The hygroscopicity obtained using the FHH framework explains water uptake and droplet growth through adsorption. At the point of activation, the FHH hygroscopicity explicitly depends on the dry particle size and the corresponding critical wet diameter. That is, $\theta \longrightarrow \theta_c = \frac{D_{p,c}-D_{dry,c}}{2D_w}$ at the point of activation. $\kappa_{FHH}$ in Eq. (14) can be further simplified $(\kappa_{FHH,s})$ such that at the point of activation,

$$\kappa_{FHH,s} = \frac{6\theta_c D_w}{D_{dry,c}} \cdot \left( A_{FHH}\theta_c^{-B_{FHH}+1} \right) \tag{15}$$

Eq. (15) can be constrained using the critical surface coverage. At the point of activation, the critical surface coverage is determined as follows,

$$\left. \frac{dS}{dD_p} \right|_c = 0 \implies 1 - \frac{2\theta_c D_w}{D_{dry,c}} - \left( \frac{2AD_w}{A_{FHH}B_{FHH}D_{dry,c}^2} \right)^{0.5} \cdot \theta_c^{\frac{B_{FHH}+1}{2}} = 0 \tag{16}$$

$\theta_c$ from Eq. (16) is substituted in Eq. (15) such that $\kappa_{FHH,s} \equiv f(D_{dry,c})$, which essentially represents the theoretical $\kappa_{FHH}$. It is important to note that $\kappa_{FHH,s}$ is particle size-dependent as opposed to $\kappa_{intrinsic}$ (Eq. (9)), which is not.

### 3.4.3 HAM Hygroscopicity

Similar to KT or FHH-AT, a single hygroscopicity parameter was developed from the HAM framework $(\kappa_{HAM})$ using Eq. (7),

$$a_{w,HAM} = \left[ 1 + \kappa_{HAM}\frac{\cdot V_s}{V_w} \right]^{-1} = x_w \cdot \exp\left( -A_{FHH} \cdot \theta^{-B_{FHH}} \right) \tag{17}$$

The inclusion of the Raoult term $(x_w)$ is the main difference between Eq. (13) and (17). $\kappa_{HAM}$ is also dependent on the experimental information ($S$, or $D_{p,c}$ along with the corresponding $D_{dry,c}$) and so Eq. (17) can be accordingly rearranged to obtain HAM single hygroscopicity parameter as follows,

$$\kappa_{HAM} = \frac{6\theta D_w}{D_{dry,c}} \cdot \left[ \frac{1}{x_w \cdot \exp(-A_{FHH} \cdot \theta^{-B_{FHH}})} - 1 \right] \implies f(D_{dry,c}, D_{p,c}) \tag{18}$$

$\kappa_{HAM}$ explains water uptake and droplet growth by combining the effects of aqueous solubility and water adsorption. At the point of activation, the HAM hygroscopicity depends on the dry particle size and the corresponding critical wet diameter. That is, $\theta \longrightarrow \theta_c$ at the point of activation where $D_{p,c}$ can be computed using the generic Eq. (6) with the help of measured $D_{dry,c}$ vs. $S$. In Eq. (6), $x_w$ is calculated using solubility partitioning as explained in section 3.3. Eq. (18) is the representation of experimental hygroscopicity of the particle based on the HAM framework. Eq. (18) can be further simplified $(\kappa_{HAM,s})$ such that at the point of activation,

$$\kappa_{HAM,s} = \frac{6\theta_c D_w}{D_{dry,c}} \left( 1 - X_w \left( 1 - A_{FHH} \cdot \theta_c^{-B_{FHH}} \right) \right) \tag{19}$$





Eq. (19) is the theoretical hygroscopicity based on the HAM framework which is constrained using the surface coverage. The constraint at the point of activation is estimated from the Eq. (6) as given by the following expression,

$$\left.\frac{dS}{dD_p}\right|_c = 0 \implies \frac{d}{dD_p}\left(a_{w,HAM}\exp\left(\frac{4\sigma_{s/a}M_w}{RT\rho_w D_p}\right)\right) = 0 \tag{20}$$

Eq. (20) provides $\theta_c$ at the point of activation to substitute in Eq. (19) and hence $\kappa_{HAM,s} \equiv f(D_{dry,c})$. $A_{FHH}$ and $B_{FHH}$ are the empirically determined parameters from FHH-AT specific to the compound. Like $\kappa_{FHH}$ and $\kappa_{FHH,s}$, $\kappa_{HAM}$ and $\kappa_{HAM,s}$ are also size-dependent. However, the size-dependence in $\kappa_{HAM}$ is variable and is controlled by the aqueous solubility of the compound. An extended derivation of $\kappa_{HAM}$ is provided in the supplemental information (Section S2).

## 4  Results

### 4.1  Köhler theory application for pure and internally mixed AAAs

The critical dry diameters $(D_{dry,c})$ at supersaturations $(S)$ in the range of 0.6% − 1.6% were calculated using PyCAT 1.0. At any given supersaturation, the $D_{dry,c}$ for each sample was calculated from the size-resolved activation ratio. The CCN measurements for pure AAAs over a range of supersaturations are shown in Figure S2 (Section S3; supplemental information). The activation diameters determined for every sample at applied supersaturations were corrected using their dynamic shape factor.

The experimental setup for shape factor measurements and the shape factor dataset for AAAs and PTA-IPTA internal mixtures are shown in Section S4.1 and S4.2 (supplemental information), respectively. The size-resolved shape factors were then used to transform the measured electrical mobility diameters to their respective volume equivalent diameters (Tavakoli and Olfert (2014); Yao et al. (2020); Gohil and Asa-Awuku (2022)). The volume equivalent diameters along with their corresponding supersaturations were then used to estimate the experimental hygroscopicity based on traditional KT $(\kappa_{KT})$, for all the AAA

samples.

The activation properties of pure AAAs (PTA, IPTA and TPTA) along with their predicted $\kappa_{intrinsic}$ and $\kappa_{KT}$ are summarized in Table 2. The $S$ versus their corresponding $D_{dry,c}$ for the samples are plotted in Figure 2(a). The experimental data is represented using individual markers. The solid and dashed lines represent the KT fits using the theoretical $\kappa_{intrinsic}$. The $R^2$ scores are provided in Table 3. PTA is observed to have the best agreement with the KT prediction $(R^2 \approx 0.99)$. IPTA and

TPTA show poor agreement with traditional KT. The lack of agreement between measurements and traditional KT predictions for IPTA and TPTA can be attributed to their significantly low aqueous solubility compared to PTA (by an order of magnitude $\sim 10^2$). In addition to the predicted and measured AAA data, $(NH_4)_2SO_4$ is also shown in Figure 2 (a).

For compounds that are considered "sparingly soluble" or "effectively insoluble" (Petters and Kreidenweis (2008); Figure 3), an explicit treatment of the compound solubility can typically improve the agreement between predicted and measured ac-

tivation properties. Based on this convention, PTA would also be considered "sparingly soluble". However, our results suggest that an explicit treatment of PTA solubility is not required. Moreover, $\kappa_{intrinsic}$ is a good representation of PTA hygroscopicity. Figure 2(b) shows the traditional and solubility-limited KT fits for internal mixtures of PTA and IPTA using their $\kappa_{intrinsic}$. The traditional KT predicts the CCN activity of the mixture containing excess PTA (5:1 mass ratio). This suggests that the





mixture dominated by PTA must have an aqueous solubility closer to pure PTA and a $\kappa_{intrinsic} \approx \kappa_{KT}$ that can be obtained
using the ZSR approximation. The agreement between traditional KT fits and experimental data reduces as the mass fraction
of IPTA increases in the mixture.

The application of solubility limited (modified) KT showed poor agreement with the pure AAAs and PTA-IPTA internal
mixtures (Figure S4; supplemental information). Modified KT overpredicted the critical supersaturation for any given dry
particle size for all 6 samples. Thus, the underprediction of AAAs CCN activity is attributed to significantly low water solubility
(in the range of $10^{-5} - 10^{-3}$ vol/vol water). Furthermore, a significant droplet growth is required to facilitate $\kappa_{solubility} =$
$\kappa_{intrinsic}$ when solubility dependence is included in the hygroscopicity analysis (Figure S5; supplemental information). The
AAA solubilities are 3 or more orders of magnitude smaller compared to highly soluble species such as ammonium sulfate
(0.42 vol/vol water) or sucrose (1.26 vol/vol water). Quantitatively, the AAA droplets should grow to about 6.5, 23 and 45
times the dry particle size of PTA, IPTA and TPTA, respectively, when $\kappa_{solubility} = \kappa_{intrinsic}$. The required droplet growth is
significantly large compared to compounds like ammonium sulfate or sucrose for which the droplet growth is 1.2 and 1.5 times
the initial particle size, respectively, when $\kappa_{solubility} = \kappa_{intrinsic}$ (Figure S6; supplemental information). All of this implies
that the hygroscopicity and CCN activity of AAAs and PTA-IPTA internal mixtures is more likely a consequence of water
adsorption, and not aqueous solubility.

## 4.2   FHH-AT application for pure and internally mixed AAAs

FHH Adsorption Theory (FHH-AT) was applied for the analysis of pure and internally mixed AAAs. Figure 3 shows the
measured $S$ vs. $D_{dry,c}$ data for pure AAAs and PTA-IPTA internal mixtures. The dashed lines represent FHH-AT fits for
their respective CCN activity datasets. It should be noted that agreement for the FHH-AT can be obtained for every set of
CCN measurements since the FHH parameters are determined by applying power law fitting to the datasets. The empirically
determined FHH parameters $(A_{FHH}, B_{FHH})$ for pure compounds and internal mixtures are summarized in Table 2.

The values of $A_{FHH}$ and $B_{FHH}$ can be used to qualitatively compare the water uptake properties of the pure and in-
ternally mixed species (Kumar et al., 2009b; Hatch et al., 2019). $A_{FHH}$ dictates the attractive forces between the particle
surface and the first adsorbed monolayer of water. A larger $A_{FHH}$ implies a tendency to adsorb a higher amount of water on
the particle surface. For the pure compounds, $A_{FHH}$ decreases in the order of PTA > IPTA > TPTA (Table 2). This suggests
a declining tendency to adsorb water. Additionally, $A_{FHH}$ for the pure PTA, IPTA and TPTA decrease like their aqueous
solubilities (Table 1). For internal mixtures, $A_{FHH}$ decreases with a decreasing PTA mass fraction (5:1 > 1:1 > 1:5). This also
suggests a declining tendency to adsorb water with a decrease in PTA concentration.

$B_{FHH}$ controls the attractive forces between the particle surface and subsequently adsorbed monolayers of water. Smaller
the value of $B_{FHH}$, stronger the attractive forces over a larger radial distance from the particle surface. For the pure compounds,
$B_{FHH}$ varies in the order of IPTA > TPTA > PTA (Table 2). This suggests that the attractive force across the adsorbed
monolayers is lowest in case of the droplets formed on IPTA particles. For internal mixtures, $B_{FHH}$ follows a similar trend as
$A_{FHH}$ and decreases with a decreasing PTA mass fraction (5:1 > 1:1 > 1:5). This suggests that the attractive force across the
adsorbed monolayers become stronger with a decrease in PTA concentration.



It can be inferred that $A_{FHH}$ follows the trends of solubility and is most likely controlled by functional groups and $B_{FHH}$ drives overall droplet growth across different compositions and molar volumes. The results here are consistent with
Mao et al. (*submitted*), that showed that the $A_{FHH}$ values correlated with functionalized surfaces of aerosol with the same core (polystyrene latex; PSL). This suggests that the $A_{FHH}$ values may play a more important role with compounds of similar molar volume and highlight the importance of functionalized groups and isomeric structures in determining overall droplet growth.

### 4.3 Hybrid Activity Model (HAM) application for pure and internally mixed AAAs

One of the major factors affecting droplet growth studied in this work is the aqueous solubility of the compound. AAAs and their mixtures used in this work possess approximately equal molar mass and densities, and hence equal molar volumes. Nonetheless, they differ in terms of their water uptake. Analysis shows that the differences in their water uptake behavior could arise due to the significant variation between their aqueous solubilities. Results in the previous subsection show that either KT or an adsorption theory (FHH-AT) can be applied for the CCN analysis of moderate and low aqueous solubility species,
respectively. Alternatively, the Hybrid Activity Model (HAM) that sandwiches the FHH isotherm with the Raoult's law through solubility partitioning may agree well with the experimental data.

Figure 4 shows the $S$ vs. $D_{dry,c}$ measurements for AAAs and PTA-IPTA internal mixtures plotted along with their HAM fits. The dot-dashed lines represent the HAM fits for the respective CCN dataset. The calculation of the water activity term for all the samples studied in this work was done following the method described in section 3.3. It was observed that KT,
FHH-AT and HAM provided similar fits for samples with aqueous solubility of the order of $10^{-3}$ $m^3$ $m^{-3}$. Thus similar fits for KT, FHH-AT and HAM was observed for the samples with higher PTA mass percentage (pure PTA and 5:1 PTA-IPTA mixture). The comparison of the goodness of fit between KT, FHH-AT and HAM can be made using the $R^2$ scores provided in Table 3. For pure PTA and 5:1 PTA-IPTA samples, all three models provided a goodness of fit. As the aqueous solubility of the sample was decreased (1:1 and 1:5 PTA-IPTA mixtures, pure IPTA and pure TPTA, in that order), HAM still provided
an improved CCN activity prediction for the samples ($R^2$ scores of 0.92, 0.97, 0.94, 0.91, respectively; Table 3). FHH-AT and HAM provided similar and improved $R^2$ scores along the decline in the aqueous solubility of the species, whereas the $R^2$ scores corresponding to KT fits were found to decline with decreasing aqueous solubility of the samples. Moreover, the $R^2$ scores for HAM fittings were observed to be uniformly > 0.9 and generally higher than those obtained for FHH-AT.

### 4.4 Hygroscopicity parameterization for supersaturated conditions

The $S$ vs. $D_{dry,c}$ of the AAA samples were transformed into a single hygroscopicity parameter ($\kappa$) based on KT, FHH-AT and HAM (Section 3.4). Figure 5 shows a closure plot between theoretical and experimental $\kappa$ estimated for PTA, IPTA, TPTA and PTA-IPTA internal mixtures from KT, FHH-AT and HAM. The closure analysis provides a better understanding of the applicability of different CCN models. The shaded portion of the graph denotes a 95% confidence interval across a 1-1 agreement line (dashed, black).





The theoretical $\kappa$ for KT has been represented using size-independent $\kappa_{intrinsic} = 0.172$ calculated using Eq. 4 and
Eq. 5, respectively. $\kappa_{KT}$ computed using $S$ vs. $D_{dry,c}$ measurements are plotted for each compound. For KT (solid circles),
the agreement between $\kappa_{KT}$ and $\kappa_{intrinsic}$ decreases with a decreasing aqueous solubility of the solute. Specifically, the
experimental $\kappa$ lies within 95% confidence of the theoretical $\kappa$ of pure PTA, 5:1 PTA-IPTA internal mixture, and 1:1 PTA-
IPTA internal mixture. TPTA is the sample with the lowest aqueous solubility and hence the lowest agreement between $\kappa_{KT}$
and $\kappa_{intrinsic}$.

The theoretical adsorption-based parameterization ($\kappa_{FHH,s}$) and $\kappa_{FHH}$ computed from the experimental data using the
FHH-AT framework are shown using solid diamond markers in Figure 5. The $\kappa_{FHH,s}$ and $\kappa_{FHH}$ were estimated using Eq.
(15) and (14), respectively. It was found that $\kappa_{FHH}$ had a generally good agreement with their respective $\kappa_{FHH,s}$ ($R^2$ in
the range of 0.91 to 0.99). The lowest agreement between FHH-AT $\kappa$ was observed for PTA and the 5:1 PTA-IPTA internal
mixture, as both they likely have the highest aqueous solubilities among the studied samples. Moreover, the $\kappa_{FHH}$ and $\kappa_{FHH,s}$
values of IPTA and TPTA are highly consistent with each other.

The theoretical and experimental $\kappa_{HAM}$ were computed using Eq. (19) and (18), respectively. The datapoints for $\kappa_{HAM}$
and $\kappa_{HAM,s}$ are denoted using solid squares in Figure 5. The most important feature of the HAM-based $\kappa$ framework is that
it explicitly accounts for the compound solubility within the hygroscopicity parameterization. Accounting for the contribu-
tion from the solid organic phase and dissolved aqueous phase to the overall hygroscopicity of the solute generates the best
agreement between the $\kappa_{HAM}$ and $\kappa_{HAM,s}$ values. Consequently, the $R^2$ scores observed between $\kappa_{HAM}$ and $\kappa_{HAM,s}$ of the
6 AAA samples are $> 0.97$. It is also important to note that $\kappa$ values for AAA samples obtained from FHH-AT and HAM
frameworks are smaller than those obtained using KT.

### 4.5   Droplet growth in subsaturated conditions

All the measurements shown in Figures 6 were performed at a 95% RH. Figure 6 (a-c) show the droplet sizes ($D_{wet}$) with
respect to their initial dry sizes ($D_{dry}$) for pure PTA, IPTA and TPTA. Figure 6 (d-f) show the $D_{wet}$ with respect to the $D_{dry}$ for
PTA-IPTA internal mixtures. The $D_{wet}$ predictions based on the KT-Raoult term, FHH isotherm and hybrid water activity were
derived from the parameters provided in Table 2. The Raoult model estimates (black dashed lines) for the pure and internally
mixed samples were generated using their average hygroscopic growth factor ($G_f$; Fig. 6, Eq. (7)). The supersaturated average
$\kappa$ of 0.17 for the AAA samples was used to obtain the theoretical $D_{wet}$ and $G_f$ at given dry sizes. The $R^2$ scores for the
KT-Raoult model are summarized in Table 3. The KT-Raoult model agreed well for pure PTA and 5:1 PTA-IPTA mixture.

The red dashed lines in Figure 6 show the $D_{wet}$ estimated using the FHH isotherm (Eq. (4)). The empirical FHH parame-
ters used here were determined by fitting the FHH-AT to the supersaturated CCNC measurements (Section 4.2; Table 2). FHH
noticeably underpredicts the hygroscopic behavior of the AAAs except for IPTA and TPTA in the subsaturated regime ($R^2$
estimates in Table 3). This implies that the insoluble behavior of IPTA and TPTA can be represented with high certainty in
subsaturated as well as the supersaturated regime, using the FHH theory. Moreover, the KT and FHH models (that agreed for
soluble compounds, PTA and 5:1 PTA-IPTA mixture) have different droplet growth predictions in the subsaturated regime.





The blue dashed lines in Figure 6 show the $D_{wet}$ estimated using the comprehensive hybrid water activity expressions described in Section 3.3 (Eq. (6)). Again, the hybrid water activity requires the empirical FHH parameters obtained by fitting
FHH-AT to the supersaturated CCNC measurements (Table 2) and the aqueous solubility of the compound to account for the dissolved fraction of solute (Table 1). The hybrid water activity replicated the subsaturated water uptake of all 6 of the AAAs with high certainty ($R^2$ estimates in Table 3). This is due to the explicit consideration of both compound solubility and water adsorption to describe the droplet growth process. Notably, the hybrid water activity is similar to either the KT-Raoult or the FHH isotherm depending on the compound solubility. For sparingly soluble samples (e.g., pure PTA), the KT-Raoult and
hybrid water activity generated similar fits ($R^2$ of 0.938 and 0.948, respectively). For effectively insoluble samples (e.g., pure TPTA), the FHH isotherm and hybrid activity generated similar fits ($R^2$ of 0.998 and 0.999, respectively).

The sub- and supersaturated analyses are consistent with the equilibrium curves for the pure and internally mixed AAA samples. Figure 7 shows droplet growth predicted using KT, FHH-AT and HAM corresponding to one of the experimentally determined $D_{dry,c}$. The predicted critical supersaturations ($S_c$) are also shown in the plots. KT predicted $S_c$ values deviate
significantly (> 10%) from the experimental $S_c$, as the aqueous solubility of the solute decreases. This is because KT for the structural isomers assumes similar droplet growth ($\kappa_{intrinsic}$ is ~0.17). However, FHH-AT and HAM require higher supersaturations and are less CCN active and therefore the points of activation are shifted upwards and to the left. At a given relative humidity (RH) < 100%, the KT-derived $D_{wet}$ is found to be larger than those predicted using either FHH-AT or HAM. This is consistent with the models and experimental data at 95% RH shown in Figure 6. KT-based $D_{wet}$ was found to be close
to the experimental $D_{wet}$ for pure PTA and 5:1 PTA-IPTA mixture, whereas $D_{wet}$ from FHH-AT and HAM were found close to the experimental $D_{wet}$ for the remaining solutes. After critical activation, there may be a jump from a water adsorption-driven droplet growth to one driven by complete dissolution of the solute (vertical jump in green line from blue to red). This is prominently seen in PTA but not in as evident in TPTA. Furthermore, multiple transitions are observed in internal mixtures.

VSA measured the water uptake of the three AAA compounds in the subsaturated regime. None of AAAs showed signif-
icant water uptake (with mass growth factors smaller than <1%) even at high RH (95%) (Figure 8). It should be noted that the VSA measurement uses materials in the range of $\mu$m to mm. Thus, the observed $\kappa_{HAM}$ in Eq. (19) decreases with increasing diameter and eventually approaches zero. The results across different particle measurement platforms are consistent with the hygroscopicity parameterization that is particle size-dependent.

It should be noted that in this work, we explicitly account for particle shape morphology (dynamic shape factor) and
correct the electrical mobility diameters to volume equivalent diameters as described in Gohil and Asa-Awuku (2022). Shape factors were measured and computed for all samples studied (Figure S3). Over the mobility diameters of interest (from 50nm to 150nm), the dynamic shape factor values were found to range from 1.00 to 1.08, and were therefore within 10% of 1.00. This suggests that the AAA samples studied in this work are composed mainly of spherical particles. The application of the dynamic shape factor of aerosols composed of fractals/agglomerates such as black carbon to the transition from soluble to
sparingly soluble activation must be considered in future work.





## 5 Summary and Implications

This paper presents the droplet growth analysis of AAAs using a new Hybrid Activity Model (HAM). HAM estimates the thermodynamics of the droplet growth by combining the aqueous solubility of the compound in an adsorption activation framework. HAM accounts for the contributions from undissolved as well as the dissolved fractions of the particle mass to predict droplet growth. Thus, HAM is able to predict critical properties (e.g., $D_{wet,c}$, $S_c$, $G_f$) for droplet growth in both the supersaturated and subsaturated regimes.

HAM also predicts the droplet growth of internal mixtures. The three PTA-IPTA internal mixtures (5:1, 1:1, and 1:5 with respect to PTA) show a clear transition from sparingly water soluble to effectively water insoluble mixtures (Petters and Kreidenweis, 2008). For a mixture containing two or more components, the water activity based on Raoult's law is computed using solubility partitioning (Riipinen et al., 2015). Moreover, a solubility limit of $\sim 8 \times 10^{-4}$ vol/vol water (corresponding to a 3:1 internal mixture with respect to PTA) was determined using solubility partitioning. Below this limit, the discrepancies in CCN activity will likely be > 10% for traditional KT. It is therefore reasonable to assume that the effect of adsorption on droplet growth would be more dominant in determining the growth of the pure and internally mixed AAAs as their solubilities are decreased below $\sim 8 \times 10^{-4}$ vol/vol water. Current literature considers the two paradigms separately and HAM provides a continuum to bridge and combine both mechanisms.

To do so, HAM requires three compound-specific parameters ($C_i$, $A_{FHH}$ and $B_{FHH}$) and the use of the full HAM in cloud microphysical models may extend the computational burden to account for the aerosol chemistry. Therefore, a single hygroscopicity parameter was also developed and exhibited an improved hygroscopicity parameterization for all solutes studied in this work. Raoult's law was generally overpredicts the hygroscopicity of effectively insoluble solutes. And the FHH isotherm generally underpredicts the hygroscopicity of sparingly soluble solutes. Combining the two droplet growth mechanisms in HAM provided a more robust approximation of the water uptake behavior in both subsaturated and supersaturated environments. Consequently, the experimental and simplified (theoretical) hygroscopicity estimates based on HAM ($\kappa_{HAM}$ and $\kappa_{HAM,s}$) showed the best agreement and highest goodness of fits when applied to the experimental data.

Overall, HAM is a promising new droplet growth model that can be potentially used for the analysis of any type of atmospheric compound. HAM is effective because it combines the characteristic features of the traditional KT with solubility partitioning and FHH-AT. Additionally, HAM differs from previous analytical frameworks that are based on compound solubility in that for any species using HAM, the particles are treated as completely undissolved at the start of the activation process. This is vital because other solubility limiting approaches begin with instantaneous dissolution and add the element of reduced solubility along the course of droplet growth. Indeed the approach is congruous to the concept of earlier works that explored the impact of slow dissolution (e.g., Shulman et al. 1996; Asa-Awuku and Nenes 2007) and aligns with more current findings that describe the droplet growth of viscous, amorphous or glassy-like aerosols (e.g., Altaf et al. 2018; Tandon et al. 2019; Zobrist et al. 2008; Mikhailov et al. 2009; Peng et al. 2022). In HAM, the contribution of the theorized undissolved fraction facilitates a surface until the particle fully dissolves, after which further droplet growth is controlled solely by the entire particle mass present in the aqueous phase. The HAM concept may have even more utility at lower temperatures and higher altitudes. In



general the solubility of compounds in water will likely decrease at lower temperatures; thus the role of surface adsorption on the undissolved fraction will be important to droplet growth. Additionally, solute viscosity of atmospheric compounds has been shown to have more significant effects on droplet growth at lower temperatures in the subsaturated regime (Kasparoglu et al., 2021). Rather than considering complex morphological parameters (diffusivity, viscosity, rheology), HAM simplifies the concept by considering the presence (or lack thereof) of a surface.

HAM developed in this work may improve our predictions of a wide variety of atmospherically relevant aerosols. For example, many atmospheric organic aerosols may vary significantly from each other in terms of their chemical structures and aqueous solubilities (Petters and Kreidenweis, 2008; Sullivan et al., 2009). Therefore, HAM may potentially improve the representation of hygroscopicity of organic aerosols in large-scale Global Climate Models (GCMs), hence reducing the uncertainties in the climate forcing due to the aerosol indirect effect.

*Data availability.* Data can be made available upon request.

*Author contributions.* KG and AAA designed the analysis for the supersaturated and subsaturated experimental data. KG and DR performed CCN and H-TDMA experiments to collect data. KG and CNM parameterized the FHH hygroscopicity. KG formulated HAM and parameterized the corresponding hygroscopicity. AAA conceived the idea for the study; designed and developed the experimental methodology. CP and MT collected VSA data and performed VSA analysis. All authors contributed to the writing and preparation of the manuscript.

*Competing interests.* The authors declare that they have no conflict of interest.

*Acknowledgements.* This material is based upon work supported by the National Science Foundation under Grant No. NSF: CHEM-2003927 and NSF: CHEM-1708337.



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

735



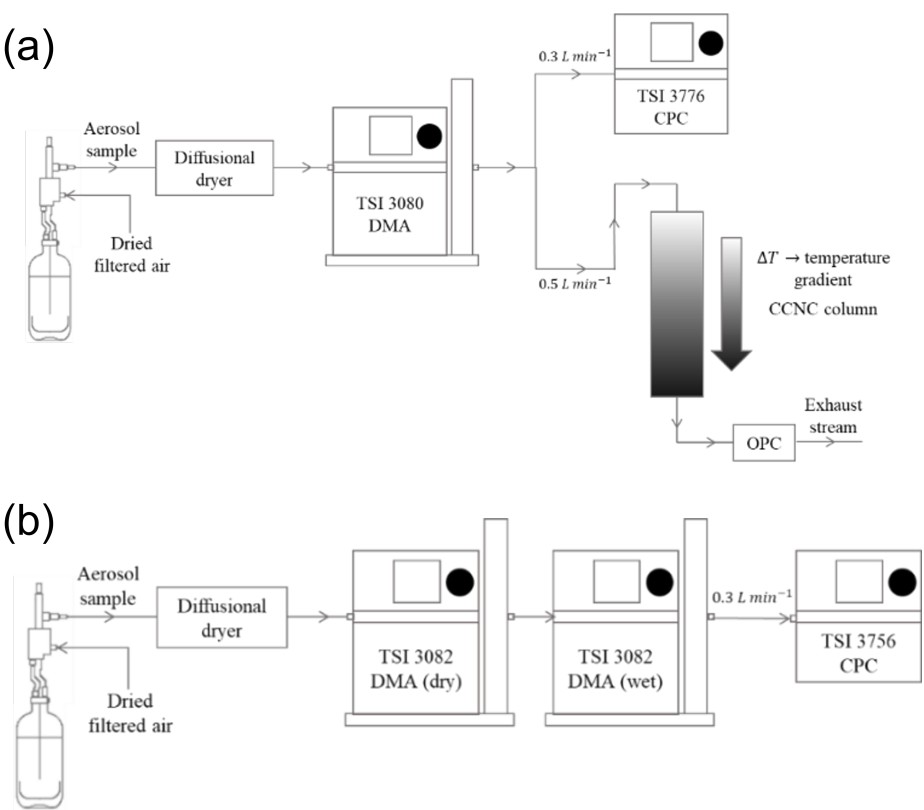

**Fig. 1.** (a) Schematic of a typical CCN measurements setup in supersaturated conditions. The DMA and the CPC collectively operate as an SMPS to obtain a distribution of dry particles. The CCNC is connected in parallel and provides the distribution of activated particles. (b) Schematic of a typical H-TDMA setup for subsaturated droplet growth measurements. The dry DMA (DMA 1) selects dry particles of a specified size. The classified particles are then humidified and passed through the wet DMA (DMA 2) and the CPC operating as an SMPS to generate the droplet distribution.



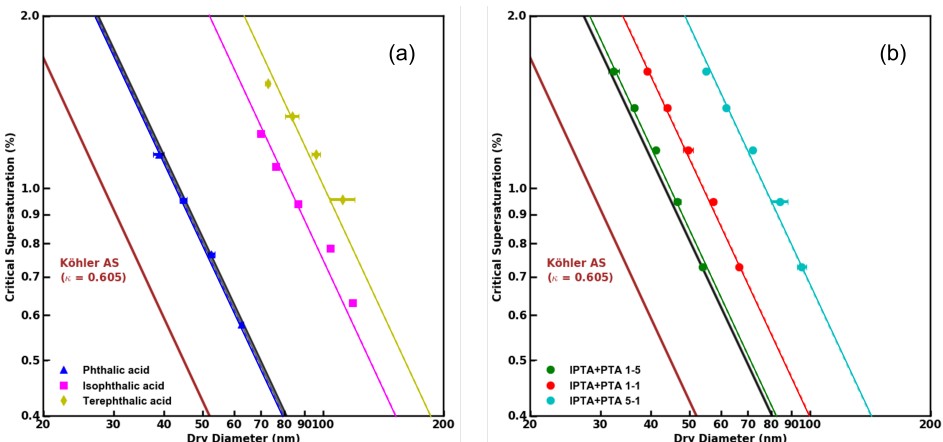

**Fig. 2.** (a) $S$ v/s $D_{dry,c}$ data obtained from supersaturated CCN measurements of pure Phthalic acid (PTA), Isophthalic acid (IPTA) and Terephthalic acid (TPTA). (b) $S$ v/s $D_{dry,c}$ data obtained from supersaturated CCN measurements of internal mixtures of PTA and IPTA. The mixtures studied shown in this plot are 5:1, 1:1 and 1:5 by mass of PTA. The solid brown line in both subplots correspond to ammonium sulfate and was used for CCNC calibration. The solid black lines were generated using the ideal Köhler theory (KT) for the respective samples, and the dashed colored lines are the KT fits obtained using the measured CCN data of each sample.

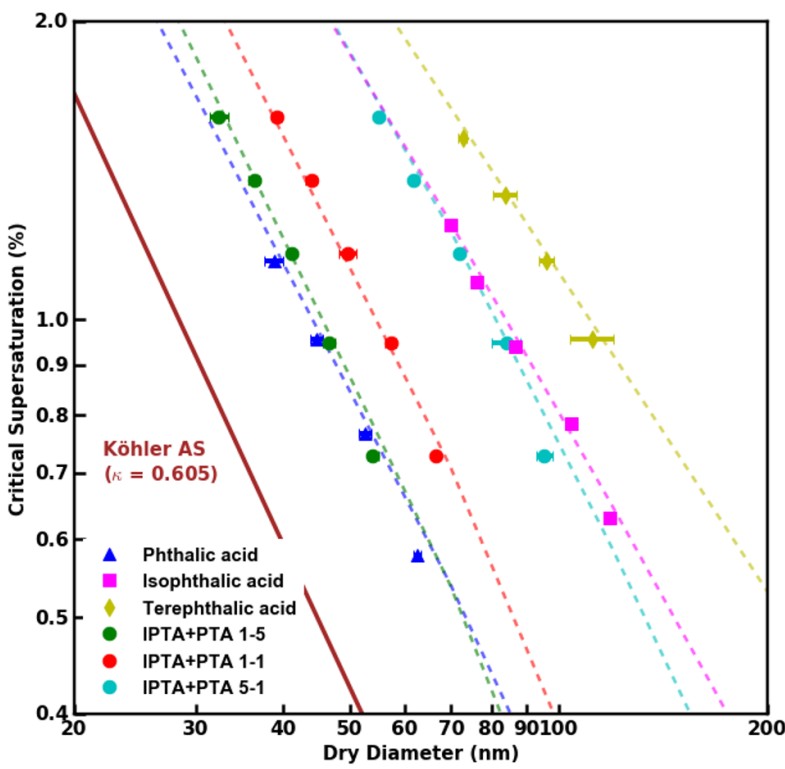

**Fig. 3.** $S$ v/s $D_{dry,c}$ data obtained from supersaturated CCN measurements of pure and internally mixed AAA samples. FHH-AT fits applied to the experimental data are shown as dashed lines.

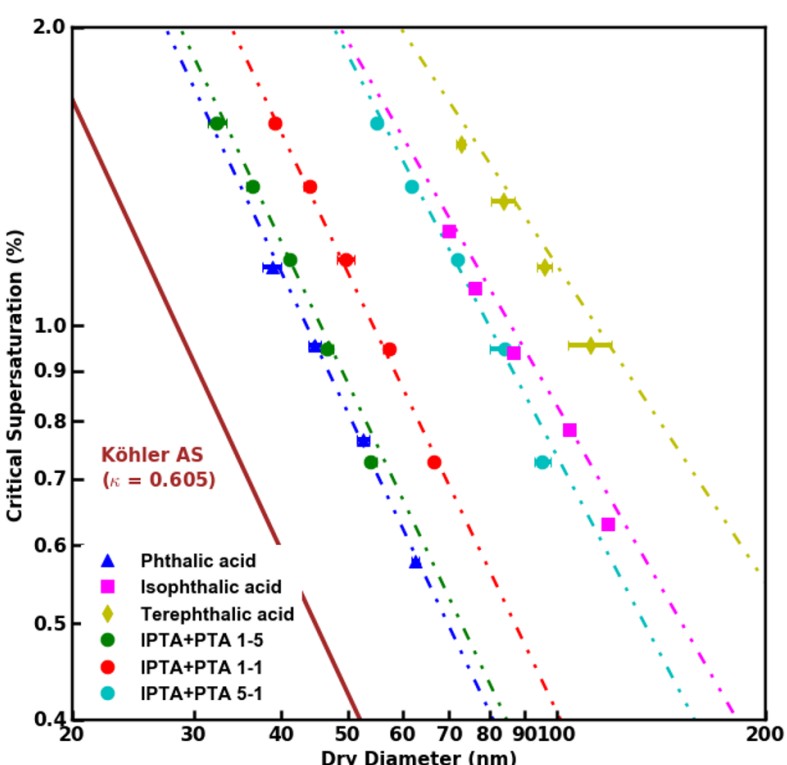

**Fig. 4.** $S$ v/s $D_{dry,c}$ data obtained from supersaturated CCN measurements of pure and internally mixed AAA samples. HAM fits applied to the experimental data are shown as dot-dashed lines.



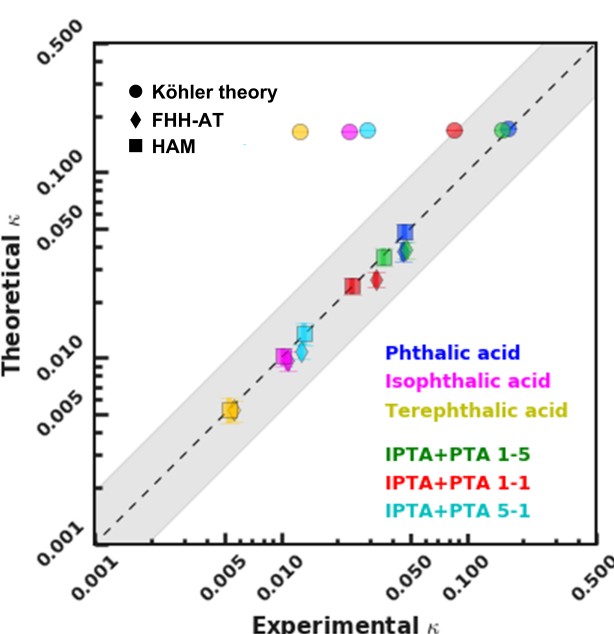

**Fig. 5.** Closure plot representing the experimental and theoretical single hygroscopicity parameters obtained using KT, FHH-AT and HAM CCN analysis frameworks. The goodness of fit was calculated for each compound and internal mixture.

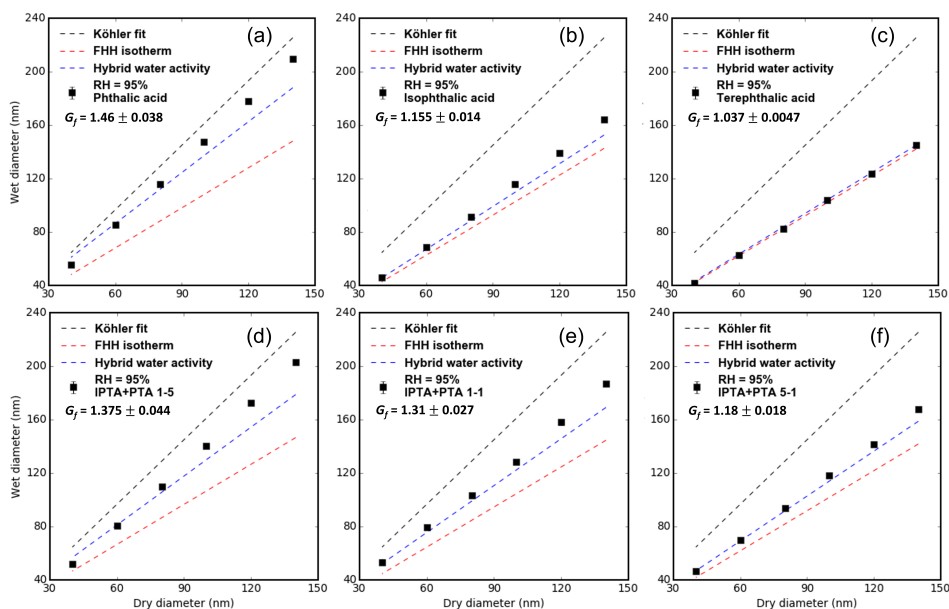

**Fig. 6.** Subsaturated measurements for pure AAA samples obtained using the H-TDMA setup are shown. (a), (b) and (c) show the $D_{wet}$ vs. $D_{dry}$ data along with model fits for pure PTA, IPTA and TPTA. (d), (e) adn (f) show the $D_{wet}$ vs. $D_{dry}$ data along with model fits for PTA-IPTA internal mixtures. The KT-Raoult term, FHH isotherm and hybrid water activity fits are shown in black, red and blue, respectively overlayed with the experimental data. The hygroscopic growth factor ($G_f$) for all AAA samples are shown in their legends.



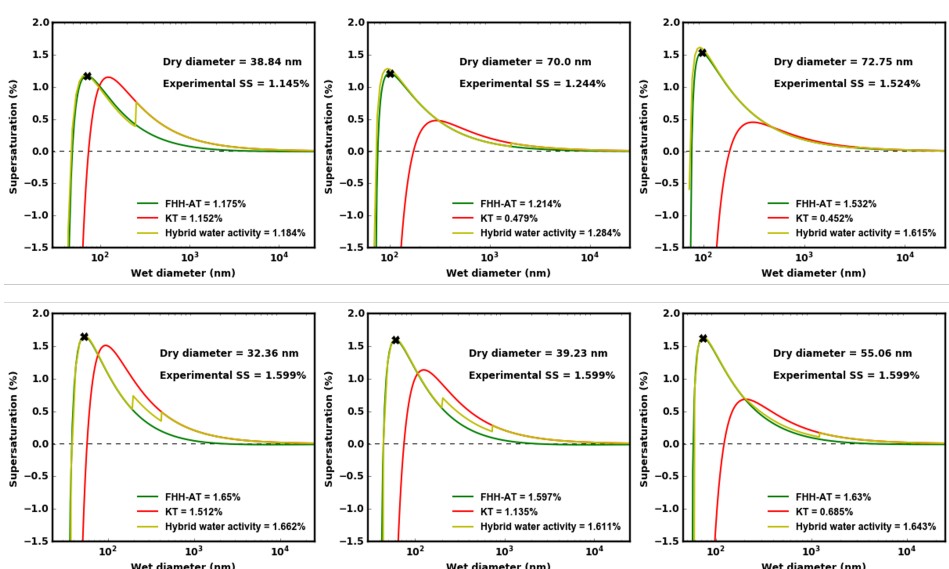

**Fig. 7.** Equilibrium droplet growth curves for PTA, IPTA, TPTA and PTA-IPTA internal mixtures are shown here. The figure header shows the solute for which the respective equilibrium curves. KT, FHH-AT and HAM lines are shown in red (solid), green (solid) and yellow (solid), respectively. An exemplary measured activation point for the respective solute is denoted using a solid red cross. The $D_{dry,c}$ and corresponding $S$ used to generate this equilibrium curves are provided in Figure 2.

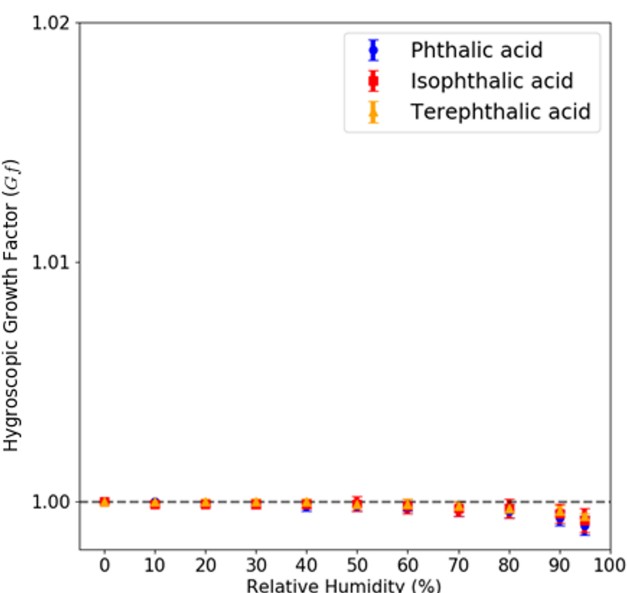

**Fig. 8.** Subsaturated measurements for pure AAA samples obtained using the VSA setup are shown. The mass hygroscopic growth factor is shown with respect to the relative humidity (RH).The measurements show that neither of PTA, IPTA or TPTA show any mass-based growth as the RH is increased from 5% to 95%.





**Table. 1.** Physical and chemical properties of AAA compounds used for calculation throughout this paper.

| Compound | Molecular weight $(M_s,\ g\,mol^{-1})$ | Density $(\rho_s,\ g\,cm^{-3})$ | Aqueous solubility $(C,\ m^3\,m^{-3})$ |
|---|---|---|---|
| Phthalic acid (PTA) | 166.14 | 1.59 | $3.77\times10^{-3}$ |
| Isophthalic acid (IPTA) | 166.14 | 1.53 | $7.84\times10^{-5}$ |
| Terephthalic acid (TPTA) | 166.13 | 1.52 | $1.12\times10^{-5}$ |



**Table. 2.** Intrinsic and experimental hygroscopicity parameter, and FHH empirical parameters used for FHH-AT and HAM analysis for pure and internally mixed AAA samples.

| Sample | Intrinsic hygroscopicity $\left(\kappa_{intrinsic} = \frac{\nu M_s \rho_w}{M_w \rho_s}\right)^a$ | Experimental hygroscopicity $\left(\kappa_{KT} = \frac{4\left(\frac{4\sigma_w M_w}{RT\rho_w}\right)^3}{27 D_{dry,c}^3 \log^2(S)}\right)^b$ | $A_{FHH}^c$ | $B_{FHH}^c$ |
|---|---|---|---|---|
| Phthalic acid (PTA) | 0.172 | 0.169 ± 0.007 | 0.41 | 0.76 |
| Isophthalic acid (IPTA) | 0.168 | 0.023 ± 0.0027 | 0.39 | 0.87 |
| Terephthalic acid (TPTA) | 0.165 | 0.013 ± 0.0018 | 0.16 | 0.84 |
| 5:1 PTA-to-IPTA | 0.171 | 0.159 ± 0.007 | 0.28 | 0.69 |
| 1:1 PTA-to-IPTA | 0.169 | 0.085 ± 0.003 | 0.21 | 0.65 |
| 1:5 PTA-to-IPTA | 0.168 | 0.029 ± 0.0024 | 0.11 | 0.61 |

a - $M_w = 18 \ g \ mol^{-1}$, $\rho_w = 1 \ g \ cm^{-3}$, $\nu = 1$

b - $\sigma_w = 0.072 \ J \ m^{-2}$, $R = 8.314 \ J \ mol^{-1} \ K^{-1}$, measured $D_{dry,c}$ v/s $S$

   $D_{dry,c}$= Measured critical dry diameter

   $S$= Supersaturation

c - Empirically determined FHH parameters from measured $D_{dry,c}$ v/s $S$ data for the given samples



**Table. 3.** Goodness of fit ($R^2$) scores for model fits applied to supersaturated and subsaturated measurements of pure and internally mixed samples.

| | Supersaturated $R^2$ | | | Subsaturated $R^2$ | | |
|---|---|---|---|---|---|---|
| Sample | KT | FHH-AT | HAM | KT | FHH-AT | HAM |
| Phthalic acid (PTA) | 0.99 | 0.87 | 0.99 | 0.938 | 0.459 | 0.948 |
| Isophthalic acid (IPTA) | -6.6 | 0.94 | 0.94 | -0.147 | 0.894 | 0.975 |
| Terephthalic acid (TPTA) | -17.4 | 0.9 | 0.91 | -1.472 | 0.998 | 0.999 |
| 5:1 PTA-to-IPTA | 0.71 | 0.91 | 0.91 | 0.869 | 0.549 | 0.933 |
| 1:1 PTA-to-IPTA | -1.5 | 0.89 | 0.92 | 0.604 | 0.667 | 0.957 |
| 1:5 PTA-to-IPTA | -5.86 | 0.94 | 0.97 | 0.007 | 0.894 | 0.987 |