# Peer review of "Hybrid Water Adsorption and Solubility Partitioning for Aerosol Hygroscopicity and Droplet Growth"

_Atmospheric Chemistry and Physics, 2022_

## Author Comment (AC1)

We thank the reviewers for their insightful comments and feedback. We have provided our point by point responses in blue text.

**Hybrid Water Adsorption and Solubility Partitioning for Aerosol Hygroscopicity and Droplet Growth**

**Responses to Referees**

**Referee 1:**

This manuscript studied the hygroscopity and CCN activity of the organic compounds with low solubility under sub-/super-saturation conditions, and the authors developed a new analytical framework, Hybrid Activity Model (HAM), to simulate the CCN activities of compounds with varying chemical structures and aqueous solubility. This article is interesting and worthy of publication.

Comment 1: Is HAM also performance good for simulating the mixture of insoluble compounds (such as black carbon) with the low solubility compounds? Give some discussion regarding that?

Response: This is a great point and a question we have already begun to address. Our future work includes looking into the application and effectiveness of HAM for the water uptake and droplet growth predictions for extremely low water solubility species, particularly black carbon. In our soon to be submitted work, we have used the Cabot Vulcan XC72R type black carbon to explore the mixtures of effectively insoluble compounds. HAM is applied to black carbon in pure state as well as to black carbon mixed with structurally isomeric compounds with varying aqueous solubility. The structural isomers from Gohil et al. (2022) are chosen to create mixtures and study their water uptake characteristics. Thus we have evidence that HAM also works well for mixtures. In this manuscript we have added a few sentences to provide additional discussion as follows:

"The next step is to evaluate the application of HAM for the CCN analysis of aerosol mixtures for a wider range in aerosol species and compositions. The shift from volume to surface based absorption principles may be more appropriate for significantly water-insoluble compounds. Specifically, the application of HAM can be examined for the hygroscopic growth and water uptake on black carbon agglomerates."

Comment 2: The SS set to 0.6-1.6% in this study, while the SS calibration with the AS only covers from 0.2-1%, what's the uncertainty of the SS above 1%?

Response: We have made a correction. The calibration of the CCNC was performed over supersaturations in the range of 0.2%-1.6%. The CCNC can perform measurements of number concentration over the a supersaturation range of 0.2%-1.6% with minimum uncertainties in the particle counts accompanied by low fluctuations in the temperature gradient across the CCNC column. The calibration plot and table have been updated in the revised supplemental information to contain the supersaturations between 0.2% and 1.6% supersaturations.

Old plot:

[Figure]

New plot:

[Figure]

**Table S1. Sample CCN Counter (CCNC) calibration data using $(NH_4)_2SO_4$**

| Supersaturation Setting (%) | Calibrated Supersaturation (%) | Critical Dry Diameter (nm) |
|---|---|---|
| 0.2 | 0.215 | 75.6 ± 2 |
| 0.3 | 0.308 | 61.7 ± 0.6 |
| 0.4 | 0.402 | 52.3 ± 0.6 |
| 0.5 | 0.493 | 45.5 ± 1 |
| 0.6 | 0.586 | 41.2 ± 0.4 |
| 0.8 | 0.771 | 34.7 ± 0.7 |
| 1.0 | 0.957 | 29.6 ± 0.6 |

New table:

**Table S1. Sample CCN Counter (CCNC) calibration data using $(NH_4)_2SO_4$**

| Supersaturation Setting (%) | Calibrated Supersaturation (%) | Critical Dry Diameter (nm) |
|---|---|---|
| 0.2 | 0.215 | 75.6 ± 2 |
| 0.4 | 0.402 | 52.3 ± 0.6 |
| 0.6 | 0.586 | 41.2 ± 0.4 |
| 0.8 | 0.771 | 34.7 ± 0.7 |
| 1.0 | 0.957 | 29.6 ± 0.6 |
| 1.2 | 1.125 | 26.1 ± 0.9 |
| 1.4 | 1.357 | 23.1 ± 1.1 |
| 1.6 | 1.546 | 21.2 ± 1. |

Comment 3: Humidifier should be added between 2 DMAs in Fig. 1(b). Can you also please specify the residence time of the particles inside the humidifier, and prove this time is enough for the low-solubility (low hygroscopicity) compounds to reach the equilibrium?

Response: This is correct - the humidifier was attached between the 2 DMAs. The nafion tube membrane used for humidification is now shown in the revised Fig. 1(b). The nafion tube was 50cm long and was calibrated using (NH4)2SO4 before measurements for any of the pure or mixed aromatic acid sample was performed. It is estimated that particles are exposed to elevated RH for ~30 seconds, longer than typical droplet formation and sufficient to establish equilibrium.

Secondly, it should also be noted that the FHH empirical parameters derived from the supersaturated measurements were applied for the hygroscopic growth analysis of the H-TDMA data obtained in the subsatured regime. This means that in the process of growth under the supersaturated conditions is consistent with the predicted growth and measurement extended to the subsatured regime. The implication here is that empirical parameters obtained from the supersaturated measurements consider equilibrium droplet growth, and should be able to predict deliquescence under subsaturated conditions when equilibrium is reached.

**Referee 2:**

The authors have developed a new single-parameter equation that is demonstrated to predict particle hygroscopic growth from sub- through super-saturated (CCN activation) ambient conditions. As such, it can be used to close the "hygroscopicity gap" that has been documented in the literature for organic compounds and mixtures, namely, the observation of very low water uptake under subsaturated conditions, implying very low hygroscopicity, but much larger implied hygroscopic growth from measured critical supersaturations required to activate into a droplet. This was achieved by conceptualizing water uptake as a two-step process as relative humidity is increased: initially, the water uptake thermodynamics are described using adsorption theory (using the FHH model); when sufficient water has been incorporated into the droplet to reach the solubility limit of the compound, then the water content of the dissolved phase is treated as obeying Raoult's Law. The model interpolates to fill in between these extremes. Interestingly, during the phase transition, the remaining solid phase is allowed to contribute additional water according to FHH predictions until full dissolution occurs.

The authors present a nice series of laboratory studies that demonstrate the fitting procedures and the superior performance of the combined model in predicting the full spectrum of water uptake. For their experimental studies, they have cleverly chosen to use three compounds with identical molecular weights, but different structures and hence solubilities. Since standard Koehler theory assumes similar behaviors based on molecular weight, the contrasts between compounds can illustrate the impacts of solubility on water uptake behaviors.

Finally, the combined model is reduced to a useful single-parameter estimate that can be fit using experimental data and that parallels prior simplified treatments for ambient aerosol hygroscopic behavior. This is a very nice feature of this work.

Overall, this paper is an interesting read and presents stimulating ideas for furthering discussion of mechanisms of water uptake in atmospheric aerosols, and how to adequately parameterize these for inclusion in models on a variety of scales.

Comment 1: A nice review of relevant literature is provided in the manuscript, including Riipinen et al. (2015). That reference seems especially relevant in setting up the theoretical equilibria (Figure 1 in Riipinen et al.) that also apply to the cases considered in the manuscript. I am not entirely clear how a corresponding model for the mechanisms proposed here would modify this figure – can the authors please comment? Are there more than two phases conceptualized in the aerosol – undissolved organic, one associated with the surface and one that is an aqueous solution of organic(s)?

Response: This is an excellent point! Figure 1 in Riipinen et al. (2015) provides a great insight into what would the two separate insoluble and soluble phases look like during water uptake. HAM was formulated on a similar assumption of the two phases. However, in contrast to the Riipinen et al. where the water uptake was controlled only by the solute fraction present in the aqueous phase, HAM also accounts for the contribution of the undissolved phase on water uptake. HAM does this with the help of FHH isotherm. The FHH isotherm computes the water vapor pressure caused due to the undissolved dry particle at the center of the droplet. The undissolved particle continues to shrink as a result of infinitesimal dissolution of the particle with the increase in droplet size.

Comment 2: Following on point 1 above, the theoretical basis for equation 6 (including equation 6b) was not clear to me. Is the assumption that the total water content is the sum of each independent water content (one from dissolution, one from adsorption)? The authors propose that aw_HAM = aw_KT x aw_FHH. How was this derived?

Response: The formulation of the 'hybrid water activity' term was derived based on the method that has been used in past studies. The net water activity term (aw) was composed of all the individual components that contribute to it. More specifically, the formulation of aw was done analogous to the way presented in Kumar et al. (2011) where the net aw = xw x f($\Theta$), where xw is the Kohler term, and f($\Theta$) is the FHH term dependent on $\Theta$. In our definition, we have used different notations for similar quantities - the difference being that our definitions of the 2 terms vary with the continued droplet growth, depending on the aqueous solubility of the sample.

Comment 3: As discussed in the manuscript, Shulman et al. (1996) early on suggested that partial dissolution of sparingly soluble compounds could generate more complex shapes for the curves expressing the equilibrium between ambient relative humidity and the particle water content. It is good that this possibility was explored here, although somewhat surprising that agreement was not better. Considering that the experiments showed the measured CCN activity to indicate higher hygroscopicity than expected (Figure S4), is it possible that small levels of impurities could affect the results? The sensitivity has been discussed previously, eg by Bilde and Svenningsson (2004), among others. Note that Hori et al. (2003) also measured CCN activity of

phthalic acid and found it to exhibit higher hygroscopicity than expected from its limited solubility. They tentatively attributed this to incomplete drying of the particles.

Response: We do not believe that the discrepancy with solubility limited Kohler theory is due to impurities. As our work shows, discrepancy can be reconciled across three different compounds if one considers contributions from water adsorption in addition to solubility partitioning. The second notion of incomplete drying of particles also is aligned with the idea that a small amount of adsorbed water will change the perceived hygroscopicity. Indeed, if a future work can revisit all previous works that have discussed discrepancies and apply this new HAM model, it may show that the HAM logic model will explain differences in sub-saturated and supersaturated experimental results.

Comment 4: Surface tension plays a role in CCN activation; less so in modulating growth factors. Although Shulman et al. (1996) found that phthalic acid did not affect the water surface tension, it seems worthwhile to at least mention the possibility of surface tension affecting CCN activity and discuss whether it applies in the systems studied, as it has been raised as one of the potential drivers of the "hygroscopicity gap" (Wex et al., 2009). In agreement with the behavior reported in this manuscript, those authors also showed low hygroscopic growth factors up to 95% relative humidity (RH), but were able to map out rapid changes in water uptake (corresponding to enhanced hygroscopicity) as RH was increased above 98%.

Response: This is a good point. The main reason why surface tension was not incorporated in our analysis is because we were more interested in looking at a somewhat isolated effect of the aqueous solubility of the compounds and their mixtures on activation and droplet growth. In order to find the right tradeoff between complexity and efficacy of the single hygroscopicity parameter, we chose to explicitly focus on the solubility of the compounds. Surface tension does contribute to the overall water activity as well as solubility partitioning, and hence should be considered in the analysis. We have added some discussion on the importance of surface tension in the "Summary and Implication" section of the revised text:

"In addition to the factors considered in this work, surface tension can potentially play a role in both the water activity term and also in the solute partition, and should therefore be treated explicitly in the droplet growth process. Incorporation of surface tension in the analysis was beyond the scope of this work, and well-designed experiments will be required to observe whether surface tension has any contribution on the water uptake of the AAAs studied in paper. Furthermore, surface effects of a given species can be parameterized within the HAM framework and subsequently into the hygroscopicity to understand such effects for partially insoluble to effectively insoluble systems."

Comment 5: In a companion paper to Wex et al. (Petters et al., 2009), those authors described two alternative explanations for the hygroscopicity gap. These included (1) gradual dissolution of multicomponent particles and (2) nonideality of aqueous organic solutions (Amundson et al., 2007). For the discussion of (2), equation (12) in Petters et al. (2009) is analogous to equation (6) in the manuscript (the HAM model equation), where in HAM the activity coefficient has been replaced by the water activity expression according to the FHH model (equation 6b). Similar to the FHH model, the activity coefficient is based on an exponential relationship. Therefore, is it possible that HAM can fit the experimental data because, as found for models for activity coefficient, the functional form selected can effectively estimate the variation of the activity coefficient over the large range of solution compositions? Can the authors comment on any parallels in the two approaches, and whether one may have a sounder first-principles basis (e.g., surface coverage data can better constrain HAM)?

Response: The activity coefficient ($\gamma$w, Petters et al. (2009) or Riipinen et al. (2015)) is 1 for infinite dissolution and explicitly treated in the water activity term when the aqueous dissolution of the compound is solubility-limited. However, even for the case of explicit activity coefficient, the contribution of the undissolved solid phase of the compound is not considered in droplet growth. HAM accounts for a combined effect of the solubility-limited approach (solubility partitioning, as defined in Riipinen et al. (2015)) and water adsorption (FHH isotherm) for their overall contribution to droplet growth. For certain cases of solubility (we observed for > ~10^-2 vol / vol water; based on the values in Riipinen et al. (2015)), HAM and solubility limited approached yield similar results. We have added this discussion in the "Theory" section of the manuscript to clarify differences between HAM and the previous frameworks:

"Previous studies have discussed several other mathematical models built upon the traditional Kohler theory under different conditions. One such example is that of the solubility-partitioned Kohler theory (Petters et al. 2009, Riipinen et al. 2015) which explicitly includes the activity coefficient ($\gamma$w) of the aerosol compounds to estimate the water activity. $\gamma$w ~ 1 in the traditional Kohler theory only under the assumption of the infinite dilution of the aqueous phase of the droplet, which holds true for several highly soluble aerosol species. For limited water solubility compounds, $\gamma$w is calculated by treating the aqueous solubility of the compound. However, even then the contribution of the undissolved fraction of the solute to the droplet growth is not treated. Another example of modified Kohler model is the 'core-shell' model (Kumar et al. 2011) that combines the FHH isotherm and Raoult's law in a single framework to evaluate the contribution of the insoluble and soluble component of the mixture, respectively, on droplet growth. In the core-shell model, partial water solubility is not considered for any of the mixture components. HAM builds up on the concepts delineated by Kumar et al. (2011) and Riipinen et al. (2015) and considers all particles as a 'core-shell' morphology, while also treating all the components as partially water-soluble."

Comment 6: Line 295: These statements are not completely accurate. Because the kappa parameter obeys volume mixing rules, that treatment can be applied to mixtures of soluble and insoluble (but wettable) compounds. The latter are assigned kappa=0.

Response: This is true. The  mathematical expression provided in these lines will generally be applicable for computing the overall kappa of the mixture regardless of the solubility considerations of the individual components of the mixture. We have added the following additional statement in the "Theory" section of the manuscript to clarify that:

"Generally, kappa for a mixture in Eq. (10) can be applied to mixtures of soluble and insoluble compounds where the kappa of the insoluble species are considered to be 0."

Comment 7: The authors did a nice job reporting on calibrations and shape factor considerations in the Supplement, thank you for the documentation.

Response: Thank you for your insightful comments.

**References:**
Kumar, P., I. N. Sokolik, and Athanasios Nenes. "Cloud condensation nuclei activity and droplet activation kinetics of wet processed regional dust samples and minerals." Atmospheric Chemistry and Physics 11.16 (2011): 8661-8676.

Petters, M. D., Wex, H., Carrico, C. M., Hallbauer, E., Massling, A., McMeeking, G. R., ... & Stratmann, F. (2009). Towards closing the gap between hygroscopic growth and activation for secondary organic aerosol–Part 2: Theoretical approaches. Atmospheric Chemistry and Physics, 9(12), 3999-4009.

Riipinen, Ilona, Narges Rastak, and S. N. Pandis. "Connecting the solubility and CCN activation of complex organic aerosols: a theoretical study using solubility distributions." Atmospheric Chemistry and Physics 15.11 (2015): 6305-6322.